# Factors Influencing Neonatal Gut Microbiome and Health with a Focus on Necrotizing Enterocolitis

**DOI:** 10.3390/microorganisms11102528

**Published:** 2023-10-10

**Authors:** Kay D. Beharry, Magdalena Latkowska, Arwin M. Valencia, Ahreen Allana, Jatnna Soto, Charles L. Cai, Sergio Golombek, Ivan Hand, Jacob V. Aranda

**Affiliations:** 1Department of Pediatrics, Division of Neonatal-Perinatal Medicine, State University of New York Downstate Health Sciences University, Brooklyn, NY 11203, USA; magdalena.latkowska@downstate.edu (M.L.); charles.cai@downstate.edu (C.L.C.); sergio.golombek@downstate.edu (S.G.); jacob.aranda@downstate.edu (J.V.A.); 2Department of Pediatrics, Division of Neonatal-Perinatal Medicine, Saddleback Memorial Medical Center, Laguna Hills, CA 92653, USA; valwin_50501@yahoo.com; 3Department of Pediatrics, State University of New York Downstate Health Sciences University, Brooklyn, NY 11203, USA; ahreen_allana@hotmail.com (A.A.); jatnna.soto@downstate.edu (J.S.); 4Department of Pediatrics, Division of Neonatal-Perinatal Medicine, Kings County Hospital Center, Brooklyn, NY 11203, USA; ivan.hand@nychhc.org

**Keywords:** cell death, factors influencing neonatal gut microbiome, fetal gut colonization, gut development, microbiome functions, necrotizing enterocolitis, neonatal gut colonization, oxidative stress, intermittent hypoxia

## Abstract

Maturational changes in the gut start in utero and rapidly progress after birth, with some functions becoming fully developed several months or years post birth including the acquisition of a full gut microbiome, which is made up of trillions of bacteria of thousands of species. Many factors influence the normal development of the neonatal and infantile microbiome, resulting in dysbiosis, which is associated with various interventions used for neonatal morbidities and survival. Extremely low gestational age neonates (<28 weeks’ gestation) frequently experience recurring arterial oxygen desaturations, or apneas, during the first few weeks of life. Apnea, or the cessation of breathing lasting 15–20 s or more, occurs due to immature respiratory control and is commonly associated with intermittent hypoxia (IH). Chronic IH induces oxygen radical diseases of the neonate, including necrotizing enterocolitis (NEC), the most common and devastating gastrointestinal disease in preterm infants. NEC is associated with an immature intestinal structure and function and involves dysbiosis of the gut microbiome, inflammation, and necrosis of the intestinal mucosal layer. This review describes the factors that influence the neonatal gut microbiome and dysbiosis, which predispose preterm infants to NEC. Current and future management and therapies, including the avoidance of dysbiosis, the use of a human milk diet, probiotics, prebiotics, synbiotics, restricted antibiotics, and fecal transplantation, for the prevention of NEC and the promotion of a healthy gut microbiome are also reviewed. Interventions directed at boosting endogenous and/or exogenous antioxidant supplementation may not only help with prevention, but may also lessen the severity or shorten the course of the disease.

## 1. Introduction

The microbiome is a diverse community of microorganisms consisting of bacteria, archaea, fungi, algae, and small protists that inhabit the bodies of mammals. They play key roles in energy homeostasis, metabolism, gut and immune health, and neurodevelopment [1,2]. The inclusion of viruses into the “microbiome” definition is controversial because they are not usually considered as “living” microorganisms [2]. The microbiome can be found on the skin surface, in the intestinal tract, lungs, and many other organs, with the most abundant species being bacteria [3], and is highly influenced by several factors including diet, the environment, medical interventions, and disease states [1]. In this regard, the human microbiome is now considered to be an organ [4]. The most studied microbiome is that of the intestinal tract (gut), which has a greater degree of diversity than the microbiome of other body sites. A healthy gut microbiome influences the brain, liver, and lung, emphasizing its importance as a central organ for human health [2,5,6,7]. In humans, early bacterial colonization with species, such as *Lactobacillus*, and other pioneering populations might occur during vaginal delivery and breastfeeding [8]. However, preterm infants born at <28 weeks’ gestation, have delayed gut colonization, and many factors, including oxygen therapy, mechanical ventilation, intermittent hypoxia (IH), antibiotics, and reduced breast milk intake, cause disruptions or alterations in gut development with an increased prevalence of Proteobacteria, Firmicutes, and Enterobacteriaceae (*Escherichia coli*, *Klebsiella*, *Staphylococcus*, *Propionibacterium*, and *Corynebacterium*), resulting in long-term effects on the neonate’s health, including an increased susceptibility to necrotizing enterocolitis (NEC), which is the most devastating, life-threatening gastrointestinal disease of preterm infants [9]. The goal of this review is to provide an in-depth summary of the factors that influence neonatal gut colonization and health that may lead to dysbiosis and NEC in preterm infants. These include the maternal microbiome, mode of delivery, NICU environment, feeding mode, oxygen therapy, antibiotics, and IH during the first few weeks of life. Current and future therapies, including the use of a human milk diet, probiotics, prebiotics, synbiotics, and fecal transplants, as potential treatments for the prevention of NEC in preterm infants, and the avoidance of disruption as well as the promotion of a healthy gut microbiome are also reviewed.

## 2. Gut Development

Gut development occurs in five major phases: (1) Phase 1 (embryonic) begins immediately after conception and continues until the fifth week of gestation. (2) Phase 2 is characterized by a rapid growth and formation of villi. Phase 3 (late gestational age) is the preparatory stage for extra-uterine life, and it is when intestinal cells actively divide and migrate up the villus to the tips and crypts of the villi. The forming villi are separated by a proliferating intervillus epithelium, which is shaped downward to form crypts [10,11]. Phase 4 (neonatal) begins after birth with rapid mucosal differentiation and development once the infant receives enteral feeding. Phase 5 (weaning) is the final phase of gut development and occurs in early childhood during the transition to solid food [12].

### 2.1. The Intestinal Barrier

The intestinal barrier is composed of multiple layers: the commensal microbiota layer, outer mucus layer, and the intestinal epithelial layer. The mucus layer is the first physical barrier that bacteria meet in the digestive tract. It protects the epithelium from harmful microorganisms and antigens, but also acts as a lubricating agent for intestinal motility. The epithelial layer is a single layer comprising five different cell types: enterocytes, endocrine cells, M cells, goblet cells, and Paneth cells [13]. These cells form an important physical and biochemical barrier that separates and prevents the microbial contents in the lumen from entering the body [14]. The microbiota layer is composed of 100 trillion microorganisms [15]. It is responsible for protection and metabolic and structural functions, and metabolizes undigested dietary products [16].

### 2.2. Cells of the Intestinal Epithelium

The crypts of Lieberkühn contains all stem and proliferating cells in the intestinal epithelium. The intestinal crypts and villi develop in human fetuses from 8–24 weeks’ gestation. The crypt depth and villus height increases as a function of the gestational age [17]. Proliferating cells appear at 9 weeks’ gestation and make their way down to the base of the crypts. These cells are classified according to their function, namely, absorptive (enterocytes) or secretory cells (mucus-secreting goblet, antimicrobial peptide-secreting Paneth, hormone-secreting enteroendocrine cells, and chemosensing/immunomodulatory cytokine-secreting tuft cells). **Enterocytes** are the most abundant epithelial cells in the intestine, and their primary function is to absorb and export nutrients, as well as to serve as the primary barrier for transport between the lumen and circulation [18,19]. **Enteroendocrine cells**: Enteroendocrine cells are primarily known to produce hormones in the gut in response to nutritional signals, which subsequently aid in digestion and metabolism [20]. **Goblet cells** are the most abundant secretory cell type in the intestinal epithelium. They produce and secrete mucus to provide the epithelial cells with a protective shield against noxious luminal contents. They also secrete antimicrobial peptides (AMPs), and like Paneth cells, they increase with advancing gestation, where they remain stable between 17 weeks and term gestation. **Paneth cells:** Maturing Paneth cells migrate downward into the crypts, where they reside for 3–6 weeks [21]. Paneth cells mature and become abundant after term birth and migrate toward the base of the crypts, where they complete their maturation [22,23]. They secrete AMPs into the lumen, which serve as an initial line of defense against potentially harmful pathogens in the gut [24]. AMPs produced by Paneth cells are natural antibiotics. There are two major categories, α-defensins and β-defensins, and of these, there are six α-defensins (DEFA 1–6) that exert their effects on pathogens but are harmless to non-pathogenic commensal microorganisms [25]. Paneth cells do not appear in the intestine until approximately halfway through intestinal development and maturation (22–24 weeks of human gestation), and it takes months before they reach their optimal density and become fully functional [22]. In this regard, premature infants are born before Paneth cells are fully developed and fully functional, which is an important developmental factor that may contribute to NEC [22,26,27]. **Tuft cells**: Tuft cells, also called brush cells, are involved in the detection of taste and chemical sensations of luminal contents [28]. **M cells:** M cells are a specialized type of epithelial cells. They serve as a route of transfer for microbial products from the intestinal lumen to antigen-presenting cells. M cells appear as early as 17 weeks [17]. **Tight junctions**: The gut epithelium is impermeable to hydrophilic solutes. Molecules and nutrients can only pass through it with the aid of specific transporters. This is due to a group of proteins that form tight junctions that seal spaces between the epithelial cells [29]. Tight junctions are initially observed at approximately 10 weeks’ gestation [17] and they play a key role in the integrity of the intestinal barrier [30]. They are made of multi-protein complexes of four classes of transmembrane proteins: occludin, claudins, junctional adhesion molecules, and tricellulin [29]. Disruptions in the tight junction integrity lead to barrier dysfunction, leaky gut, and many diseases [31]. Epithelial cells frequently undergo death as a process of renewing and maintaining tissue homeostasis; the elimination of superfluous, damaged, or aged cells; as well as in response to acute and chronic injury, such as NEC.

## 3. Stages of Gut Colonization

Gut colonization occurs in several stages, possibly beginning prenatally and maturing by three years of age. The issue of whether gut colonization begins in utero has not been fully resolved. While some studies reported the presence of microbial populations in fetal tissues [32,33,34], other studies did not find evidence of a fetal microbiome [35]. Kennedy et al. [36] concluded that a low microbial biomass and contamination were responsible for the erroneous findings. Nevertheless, studies show that many prenatal factors influence gut microbiome development and suggest that the placental microbiome may be established by the maternal oral microbiota [37,38]. Meconium (first stool) samples from 21 healthy neonates were shown to contain gut founder populations of *Staphylococcus* and *Bifidobacterium* [39]. The microbiome of newborn infants has a lower diversity compared to that of adults, and it is influenced by many factors including the duration of gestation and mode of delivery (vaginal or Cesarean section, or C-section). The newborn infant is exposed to microorganisms from their mother and the surrounding environment. Studies show a predominance of Firmicutes, Proteobacteria, and Actinobacteria with lower levels of *Bacteroidetes*, which is the dominant phylum in adults [40]. Other studies show that infants who are born vaginally acquire bacteria resembling the maternal vaginal microbiome, which is predominantly *Lactobacillus* and *Prevotella*, whereas infants who are born via C-section acquire bacteria resembling the skin microbiome (predominantly *Staphylococcus*) [41]. These pioneering microorganisms influence subsequent gut microbiome development. The microbes that infants are first exposed to at birth are thought to play roles in the subsequent maturation of microbial communities, specifically in the gut [42]. The first few years of postnatal life represent a critical time for early childhood development, and in the first year of life, the infant microbiome undergoes significant fluctuation and maturation [40]. By three years of age, the infant microbiome resembles that of an adult. The first colonizers of the infant gut microbiota are typically facultative anaerobes, followed by the accumulation of obligate anaerobes, including *Bifidobacterium*, *Bacteroides*, and *Clostridium*, during the following 6 months of age [43,44,45]. One of the most important factors that influence infant gut microbial diversity and development is diet [46]. Breast milk has a high number of prebiotics, or human milk oligosaccharides (HMOs), and the infant gut microbiome is enriched with genes involved in their digestion such as *Bifidobacteria*, *Bacteroides*, and *Lactobacillus* [47]. The digestion of HMOs produces short-chain fatty acids (SCFAs) such as acetate, propionate, and butyrate, which can be used as energy sources and lower the luminal pH to inhibit the colonization of pathogens [48]. Compared to breastfed infants, the gut microbiome of formula-fed infants was shown to be dominated by Firmicutes (*Staphylococcus*, *Streptococcus*, *Enterococcus*, *Lactobacillus*, and Clostridium), Bacteroidetes (*Bacteroides*), Proteobacteria (*Enterobacteria*), and Actinobacteria (*Atopobium*) [49]. After the introduction of solid foods, the infant gut microbiome is enriched in genes involved in the digestion of polysaccharides. These include *Bacteroides*, and the differences between breastfed and formula-fed infants decrease as the infant microbiome becomes consistent with that of an adult [44].

## 4. Fetal Gut Colonization

Although the maturation of the gut starts in utero, it matures after birth, with some functions becoming fully developed several months or years post birth [50]. Until recently, it was believed that the GI tract of the fetus was sterile, and that gut colonization started during the delivery process. Using advanced shotgun metagenomic sequencing techniques, researchers have demonstrated the presence of commensal bacteria in the amniotic fluid, uterus, placenta, and meconium [37,51,52,53,54,55,56], suggesting that the fetal gut and the intrauterine environment are not sterile, and that there is a maternal-to-fetal exchange of microbes, which highlight the importance of the maternal microbial environment to modulate the infant’s immune system [57]. By comparing the amniotic, placental, and meconium microbiota of infants delivered via C-section, Collado et al. [58] found shared features between the microbiota detected in the placenta, amniotic fluid, and infant meconium. Collado et al. also found that Proteobacteria was the most prevalent phylum in the amniotic fluid and placenta, with *Enterobacter*, *Escherichia*, and Shigella being the most predominant Proteobacteria phyla with a lower abundance in the colostrum, meconium, and infant feces. The exact mechanism of how these microorganisms pass from the mother to the fetus is unknown, but studies show a remarkable similarity between the placental and maternal oral microbiome [37]. This may suggest that the bacteria in the buccal cavity may be a key source of bacterial translocation to the placenta. The Human Oral Microbiome Database showed that the oral cavity, a major gateway into the human body, contains over 600 taxa in 13 phyla, including *Actinobacteria*, *Bacteroidetes*, *Chlamydiae*, *Chloroflexi*, *Euryarchaeota*, *Firmicutes*, *Fusobacteriia*, *Proteobacteria*, *Spirochaetes*, *Synergistetes*, and *Tenericutes* [59]. Studies show that the oral microbial diversity remains relatively stable during pregnancy, but the composition of the microbiome can undergo a pathogenic hormonal shift [60], which may be associated with many adverse pregnancy outcomes [61]. In a healthy placenta and uterus, there is an abundance of commensal bacteria [37,53], the most prevalent of which is Proteobacteria, with *Escherichia coli* (*E. coli*) being the most prevalent single species [62]. A systematic review of studies reporting on the placental microbiome found that *Lactobacillus* was the most commonly found genus in the placenta [63], which was also the main genera found in breast milk [64] and was associated with healthy gut and vaginal microbiomes [65,66]. The uterine microbiome appears to originate from the vagina, which has an abundance of *Lactobacillus* [67,68]. Other abundant microorganisms found in the uterus belong to the Firmicutes, Bacteriodetes, Proteobacteria, and Actinobacteria phyla [69]. Despite these numerous reports of possible fetal colonization, recent reports did not find evidence of a fetal microbiome, thus challenging those previous findings [36]. Whether gut colonization begins in utero or not has not been fully resolved.

## 5. Neonatal Gut Colonization

The microbial gut community is important for maintaining health, programming our immune system, and developing the intestinal tract and metabolism. It comprises all microorganisms including bacteria, viruses, and fungi and it is estimated that greater than 100 trillion commensal and non-pathogenic microorganisms inhabit the gut. Most of the microorganisms in a human infant are restricted to five dominant phyla, Firmicutes, Bacteroidetes, Actinobacteria, Proteobacteria, and Verrucomicrobiae, which provide beneficial effects. Any alterations or aberrations in their compositions during neonatal life are associated with many pediatric illnesses and adult-onset diseases [70]. The postnatal colonization of the gut generally occurs immediately during the birth process. An examination of meconium, the first fecal material from newborns, showed a predominance of *Bifidobacterium*, *Enterobacteriaceae*, *Enterococcaceae*, *Bacteroides*, and Prevotella [71]. Other studies showed that *Proteobacteria* (*E. coli*, *Klebsiella*) and *Bacilli* (*Enterococcus*, *Staphylococcus*, and *Streptococcus)* were more common [72,73,74]. In the cord blood of healthy newborns delivered via C-section, *enterococcus faecium*, *propionibacterium acnes*, *staphylococcus epidermidis*, and *streptococcus sanguinis* were reported [55]. The neonatal gut is initially dominated by *Bifidobacterium*, *Veillonella*, *Streptococcus*, *Citrobacter*, *Escherichia*, *Bacteroides*, and *Clostridium*, which are also abundant in the gut microbiota of adults [71,75], but it is eventually populated by two dominant groups of strict anaerobic bacteria belonging to the Firmicutes and Bacteroidetes phyla [76].

## 6. Functions of the Microbiome

The gut microbiome affects many aspects of our health and physiology, and its appropriate composition plays an essential role in the proper functioning of our bodies. One of the crucial functions of the microbiome is its involvement in the metabolism of indigestible carbohydrates and HMOs that escaped proximal digestion by producing three primary SCFAs, acetate, propionate, and butyrate [77]. Butyrate is a key energy source for human colon epithelial cells [78]. Its consumption improves the integrity of intestinal epithelial cells by promoting tight junctions, cell proliferation, and increasing mucin production by Goblet cells [79]. It also has potential anti-cancer properties by causing the apoptosis of colon cells. Propionate is metabolized by the liver, which participates in gluconeogenesis and decreases hepatic glucose production, subsequently reducing adiposity [78]. Intestinal bacteria also play essential roles in the biosynthesis of vitamin K and some components of vitamin B (i.e., thiamin, riboflavin, niacin, biotin, and folate). Another function of the gut microbiota is bile acid metabolism. Bacteria such as *Bacteroides intestinalis*, *Bacteroides fragilis*, and *E. coli* can convert primary bile acids into secondary bile acids via deconjugation and dihydroxylation [80]. The microbiome affects the host immune system in many ways. It affects the development and function of the innate and adaptive immune system. All three SFAs have anti-inflammatory properties [81]. SCFAs have essential roles during pregnancy and in the fetal immune system; their production is increased, which is essential for T-cells to differentiate in the thymus of the fetus [82]. They are responsible for maintaining homeostasis between the production of anti-inflammatory (IL-10) and proinflammatory cytokines (IL-8, IL-1). Changes in the microbiota composition and diversity can affect the accumulation and differentiation of lymphoid tissue in the digestive tract. Gut bacteria can promote Th17 cell production against extracellular pathogens. The gut microbiota also induces the synthesis of AMPs by the Paneth cells via a PRR-mediated mechanism. In addition, gut bacteria are involved in other signaling pathways that are important in maintaining the immune homeostasis of the intestinal mucosal barrier [83]. Lately, the relationship between the composition of the gut microbiome and the occurrence of specific diseases and behaviors has been emphasized. Recent research has found a strong connection between elevated levels of SCFAs, obesity, and metabolic changes. Obese individuals are found to have increased levels of colonic SFAs, which can result in increased adipose tissue deposition [84]. The colonization with *clostridioides difficile* (*C. difficile*), a species of Firmicutes, in early infancy has been associated with eczema, wheezing, and the development of allergic sensitization [85,86]. Recent studies have established a link between gut microbiota and behavior during the early stages of life. Research indicates that diverse gut microbiota can positively impact an infant’s cognitive ability. Overall, the gut microbiota can influence an individual’s behavior and social tendencies [82].

## 7. Factors Influencing Neonatal Gut Microbiome

### 7.1. Maternal

Several maternal factors influence fetal and infant gut colonization [87], including maternal health [88], maternal diet [89], vaginal health [90], smoking [91], and antibiotic use [92,93]. The TEDDY study, which involved six institutions in the United States and Europe and 12,500 stool samples from over 900 infants showed that other factors, such as the mode of delivery, breastfeeding, geographical location, living with siblings and furry pets, antibiotic treatment, and assisted reproductive technology, significantly influence neonatal gut colonization [94].

### 7.2. Delivery Mode

The infant gut microbiota is greatly influenced by the mode of delivery. Infants who are delivered vaginally inherit intestinal bacteria from their mother’s birth canal, including *Lactobacillus* and *Prevotella* species. In contrast, babies delivered via C-section are colonized by different bacteria, including *Clostridium*, *Staphylococcus*, *Propionibacterium*, and *Corynebacterium*, and have lower levels of anaerobic bacteria like *Bacteroides* and *Bifidobacterium* [95]. Infants delivered via C-section are colonized with microorganisms that populate the mother’s skin such as *Staphylococcus* [8]. A large study involving approximately 600 infants found that the influence of the delivery mode was the most important factor influencing the infant gut microbiota composition [96]. Infants who are born via C-section also show a reduced complexity of the gut microbiota with low abundances of *Bifidobacterium* and *Bacteroides*, which are important commensal microorganisms [97]. C-section delivery was also associated with a relative abundance of pathogenic bacteria that are common in hospital surfaces [98], which are associated with an increased risk of immune disorders such as asthma, allergy, type 1 diabetes, and obesity [71,82]. A systematic review confirmed low diversity during the first week of life in infants delivered via C-section, with low diversity in the Actinobacteria and Bacteriodetes phyla compared to vaginally delivered infants during the first 3 months of life. Similar to other reports, *Bifidobacterium* and *Bacteroides* were significantly more frequent in vaginally delivered infants, while *Clostridium* and *Lactobacillus* were more common in infants delivered via C-section [99].

### 7.3. Feeding Mode

Breast milk promotes the development of the gut microbiome by introducing probiotics and prebiotics and providing protection against pathogens. The microorganisms that are dominant in breast milk include *Bifidobacterium*, *Lactobacillus*, *Staphylococcus*, *Bacteroides*, *Enterococcus*, *Streptococcus*, and *Clostridium* [100,101,102]. Studies have documented differences in the gut microbial composition between breastfed and formula-fed infants [80]. Breast milk contains its own mix of microorganisms and non-digestible human milk oligosaccharides (HMOs) that are transferred to the baby. HMOs are classified as prebiotics and help beneficial gut bacteria grow. This growth of beneficial bacteria prevents harmful pathogens from colonizing the infant’s gut, leading to positive health effects [103]. In addition, breastfeeding provides the infant with high levels of prebiotics, fatty acids, lactoferrin, and other important nutrients that protect the infant against pathogenic infections, promotes barrier function, and stimulates immune function [104]. Breast milk contains mostly *Streptococcus*, *Staphylococcus*, *Propionibacteria*, lactic acid bacteria, and *Bifidobacterium* [100,105,106], and breastfeeding is associated with a high amount of *Bifidobacterium*, representing about 90% of the total infant microbiome in the first year [107], with a lower amount of *E. coli*, *C. difficile*, *Bacteroides*, Firmicutes, and *Lactobacilli* compared with formula-fed infants [108,109]. Exclusively breastfed infants have a lower diversity in their gut microbiome, but a higher abundance and more diverse *Bifidobacterium.* Studies show that breastfed babies tend to have more *Staphylococcus* and *Streptococcus*, while formula-fed babies have higher *Bacteroides*, *Clostridium*, *Enterobacteriaceae*, *Escherichia*, *Klebsiella*, *Enterococcus*, and *Lachnospiraceae*, with a slower colonization of *Bifidobacteria* [110]. Preterm infants who were fed with their own mothers’ breast milk had a higher diversity in their gut microbiome with a higher abundance of *Clostridiales* and *Lactobacillales* than the infants who were fed with donor milk and/or formula [111]. Korpella et al. [112] examined fecal samples from 45 breastfed preterm infants from birth to 60 days post birth and found that the microbiome developed in four phases based on the dominance of *Staphylococcus* (Phase 1 peaked between 25 and 30 weeks), Enterococcus (Phase 2 peaked between 30 and 35 weeks), Enterobacter (Phase 3 peaked at 35 weeks), and *Bifidobacterium* (Phase 4 began after 30 weeks). The authors found that the *Enterococcus* phase was only observed in the extremely premature infants and appeared to delay the microbiota succession. In comparison, formula-fed infants have more pathogenic microorganisms with a lower abundance of *Bifidobacterium* and increased *Clostridium* and *Enterobacteriaceae* (*E. coli*) [44], and they have dominances of *Staphylococci*, *Bacteroides*, *Clostridia*, *Enterococci*, *Enterobacteria*, and the genus *Atopobium* and a lower abundance of *Bifidobacterium* [113]. Over the past several years, formula composition has changed to introduce probiotics and prebiotics in an effort to more closely simulate breast milk composition. With the introduction of solid foods, the neonatal gut is exposed to more complex carbohydrates and nutrients, causing increased microbial diversity, and *Proteobacteria* and *Actinobacteria* are replaced by *Firmicutes* and *Bacteroidetes* [114], which more closely resemble the adult microbiota [45]. The changes are more pronounced in breastfed infants, with reductions in *Bifidobacteria*, *Enterobacteria*, and *Clostridium*, although *Bacteroides* is one of the most predominant microorganisms [115].

### 7.4. Environment

An infant’s gut microbiota is affected by exposure to various external environments during early development outside of the uterus [50]. Children with siblings tend to have more *Bifidobacterium* and fewer *Peptostreptococcus* bacteria. The KOALA Birth Cohort Study in the Netherlands showed that infants with older siblings had a higher number of *Bifidobacteria* and increased gut microbial diversity and richness than infants without siblings [86]. A lack of older siblings was also associated with earlier colonization by *B. adolescentis*, *Clostridium*, and *C. difficile*, while colonization with *Bifidobacteria*, *Bacteroides*, and *Lactobacillus* increased with a higher number of siblings [116].

### 7.5. Ethnicity

The gut microbiota also differs in relation to geographical location, diet, and lifestyle [81]. Studies show differences in the microbiota of children in rural Africa compared to urban Italy [117]. One study examining stool samples from 605 infants in five European countries (Sweden, Scotland, Germany, Italy, and Spain) with different lifestyles and infant feeding practices showed that children from Northern European countries had a higher proportion of *Bifidobacteria*, while higher levels of *Bacteroides* and *Lactobacilli* were found in children from Southern European countries [114]. Swedish infants have a higher colonization rate with *S. aureus* than Italian infants in their first year. During the first two weeks of life, African infants have greater prevalences of *Enterococcus* and *Lactobacillus*, and lower prevalences of *Staphylococcus* and *Bacteroides* compared to Swedish infants. Infants living in rural areas have a higher amount of *Lactobacillus* at one month old, and infants in urban areas have more *Enterococcus* during their first few months of life [110]. Similar differences were seen between children living in an urban slum in Bangladesh compared to children from an upper-middle-class suburban community in the United States [118].

### 7.6. Genetics

The gut microbiome is significantly influenced by host genetics [119,120]. Recent genome-wide association studies (GWASs) and meta-analyses showed an association between *Bifidobacterium* abundance and the lactase (LCT) gene locus [121]. Moreover, genetic variants at the ABO gene locus were significantly associated with abundances of *Bifidobacterium* and *collinsella aerofaciens*. How and when the host genetics in the growing newborn exert their influence on the neonatal and infantile microbiome and in later life stages remain to be studied.

### 7.7. Pets

A number of studies have shown that indoor household pets impact the gut microbiome of children, specifically dogs and cats [122,123]). Children who are raised with cats have a more diverse gut microbiota, with an increased population of *Peptostreptococcus* bacteria and a lower population of *Bifidobacterium* [124]. Infants who are raised in a home with pets contain animal-derived *bifidobacterium pseudolongum* compared to a pet-free home [125], lower levels of *Bifidobacteriaceae*, and a higher abundance of *Peptostreptococcaceae* [126].

### 7.8. Preterm Birth

One of the most important factors that affect the gut microbiome is the gestational age. Premature infants have a different intestinal bacterial composition compared to full-term infants [82]. In preterm infants, a number of factors influence their gut colonization that ultimately result in perturbations in the gut ecosystem or dysbiosis. Dysbiosis has been found to be associated with many diseases of the preterm infant including NEC, a devastating disease and major cause of mortality in preterm infants [9,127]. Preterm birth is often associated with sudden and rapid vaginal or C-section deliveries, resulting in a lack of maternal–infant physical interaction, less exposure of the neonate to the maternal microbiome, and a higher exposure to inflammation due to maternal infections. Studies suggest that the vaginal microbiome in pregnant women may influence preterm birth. Studies show that the vaginal microbiome changes with the gestational age, resulting in a predominance of Lactobacillus spp. [128]. However, the data from the National Institutes of Health’s integrative Human Microbiome Project (iHMP), one of the largest and most comprehensive studies of the vaginal microbiome, showed that preterm birth was associated with a low abundance of vaginal *Lactobacillus*, particularly in women of African descent [129]. The data also indicated that four taxa, *S. amnii*, BVAB1, *Prevotella* cluster 2, and TM7-H1, were all positively correlated with preterm birth and may be useful for the prediction of the risk for preterm birth. Another study reported that the vaginal microbial community in pregnancy consists largely of four species, namely, *L. crispatus*, L. iners, L. jensenii, and *G. vaginalis.* Mothers who are at risk for preterm birth often receive antibiotics, which have a significant impact on the microbiome of the fetus. An in utero exposure to pathogenic microorganisms results in the premature rupture of membranes and intrauterine infection [130]. A study involving 29 preterm infants born at 28–32 weeks’ gestation showed that female infants were more likely to have a higher abundance of *Clostridiates* and a lower abundance of *Enterobacteriales* than males [111]. An examination of 719 rectal swabs from preterm infants while in the NICU from 24 to 46 weeks demonstrated a low species diversity, with *Bacilli*, *Proteobacteria,* and *Clostridia* being the most abundant phyla, accounting for 87%, followed by *Actinobacteria* and *Bacteroidia*, which accounted for 6.5% and 5.1%, respectively [131]. Generally, the preterm microbiome is mainly made up of facultative anaerobic bacteria, including *Enterococcus*, *Staphylococcus*, *Streptococcus*, *Enterobactericaeae*, *Citrobacter*, *Enterobacter*, *Escherichia*, *Klebsiella*, *Raoultella*, *Serratia*, and *Shigella*. While aerobic bacteria are present, there are also anaerobic bacteria such as *Bacteroides*, *Clostridium*, and *Veillonella*. Beneficial gut bacteria such as *Bifidobacteria* and *Lactobacillus* that protect against harmful pathogens are not fully present until two months after birth, suggesting that the microbiome differs between preterm and term infants not just in composition but also in trajectory. Within the first six weeks, preterm infants have been observed to have decreases in *Escherichia*, *Shigella*, *Staphylococcus*, and *Prevotella*, while there are increases in *Enterobacteriaceae* (particularly *Klebsiella*), *Enterococcus*, *Streptococcus*, and *Veillonella*. NEC is linked to higher occurrences of *Enterobacteriaceae*, *Clostridium*, and coagulase-negative staphylococci. Conversely, the presence of *enterococcus faecalis* is reduced [110].

### 7.9. NICU Environment

Once admitted in the neonatal intensive care unit (NICU), the preterm infant is exposed to microorganisms that are present in the hospital environment, with limited access to maternal- and family-specific microorganisms, and greater access to NICU staff microorganisms. They often receive oxygen therapy with mechanical ventilation, intravenous catheters, enteral feeding (and possibly contaminated feeding tubes), and antibiotics, predisposing them to dysbiosis and a susceptibility to NEC [132]. A study involving the sequencing of 922 stool specimens from 58 preterm infants (≤1500 g birth weight) in the NICU showed that approximately 92% of all bacteria were Bacilli (19.3%), Proteobacteria (54%), and Clostridia (18.4%) over a 3–39-day period of life [133]. To prevent sepsis, preterm infants are frequently treated with antibiotics, which impact the first colonization of the neonatal gut. These microorganisms may include *Enterococcus*, *staphylococcus aureus*, *Klebsiella*, *Acinetobacter*, *pseudomonas aeruginosa*, and other *Enterobacteriaceae*, which are found on NICU surfaces and are the most frequent cause of nosocomial infections [134]. One study identified 794 antibiotic resistance genes in preterm infant stool samples [135]. In another study involving 2832 samples collected from 16 NICU room surfaces, hands, electronics, sink basins, and air, found that most microorganisms were associated with the skin (*Corynebacterium*), mouth (*Streptococcus*), or nose (*Staphylococcus*); with skin accounting for over 50%, followed by oral and fecal associations. The microorganisms that were found in the sink were *Rhizobiaceae*, *Pseudomonas*, *Aeromonas*, and *Enterobacteriaceae*, and the floor in front of the infant’s isolette had the highest density of microbes [136]. These environmental factors impact and reshape the preterm infant gut microbiome and immune system, making them susceptible to the development of NEC.

### 7.10. Antibiotics

Premature infants and infants that were born via C-section are more likely to receive antibiotics, which can increase their risk for future diseases like asthma, obesity, and inflammatory bowel disease. Antibiotic exposure during the perinatal period may delay the maturation of microbial activity until around 6 to 12 months after birth [50]. Antibiotics given to infants increased *Enterobacteria*, while those given to mothers during pregnancy or breastfeeding decreased *Bacteroides* and *Atopobium* clusters in babies [49]. Antibiotic exposure was linked to a decrease in both the diversity and richness of the microbiome, as well as alterations in the bacterial abundance [137], and the development of NEC [138].

### 7.11. NEC

NEC is a devastating acquired gastrointestinal disease in premature infants, afflicting about 7–11% of extremely low gestational age neonates [139,140,141], with a mortality rate of 10–30%, particularly when surgical intervention is required [142,143,144]. NEC is a multifactorial and complex disease that involves intestinal necrosis resulting from hypoxia ischemia [145]. The high mortality rate, frequent poor outcomes, and the impact on healthcare costs makes NEC one of the most serious and most expensive neonatal diseases, with the total annual cost of care in the United States being USD 500 million to 1 billion [140], and the total mean cost of care over a 5-year period is USD 1.5 million for those with short bowel resection [146]. A significantly higher rate of neurodevelopmental impairment is seen in survivors of NEC, and the age of onset of NEC is inversely related to the post-menstrual age at birth, with the risk significantly decreasing only after 34–35 weeks’ gestation [147]. A bimodal age distribution has been observed with the incidence appearing in as early as 7 days in more mature infants, less than 33 weeks’ gestation, and 2–3 weeks later for ELGANs [148,149]. The histopathologic finding of NEC is hemorrhagic–ischemic necrosis [150]. However, a subset of late preterm and term infants will present atypically, with a much more rapid progression and a more catastrophic outcome, which may or may not be preceded by a thrombotic process [151]. Several factors predispose preterm infants to NEC including inappropriate immune responses, dysbiosis and leaky gut, aggressive feeding, hyperoxia, and ROS [152].

### 7.12. Intermittent Hypoxia

Preterm infants often require supplemental oxygen therapy due to their immature lungs and respiratory system. While oxygen therapy is many times essential for the survival of premature infants, high levels of oxygen exposure can have both beneficial and detrimental effects on several systems [153], including the gut microbiome. Oxygen can lead to oxidative stress and damage the gut microbial communities. Chronic IH, as seen in adults with obstructive sleep apnea, can cause changes in the gut microbiota that will affect the hepatic and adipose tissue morphology [154]. Gut dysbiosis can be independently caused by a high-fat diet and intermittent hypoxia in mice, causing cardiometabolic disease [155]. Khalyfa et al. [156] reported that IH leads to perturbations in the gut microbiota-circulating exosome pathway, resulting in metabolic dysfunction. IH can impact the gut microbiome through multiple mechanisms. It can disrupt the gut epithelial barrier, leading to increased gut permeability and the translocation of bacteria across the intestinal wall. Additionally, it can influence the oxygen availability in the gut, favoring the growth of oxygen-tolerant bacteria and potentially altering the overall microbial composition. Studies investigating the impact of IH on the preterm gut microbiome have reported a wide variety of pathogenic microorganisms and a reduction in the abundance of commensal species, which were associated with characteristics that are consistent with NEC [157].

### 7.13. Oxidative Stress and Reactive Oxygen Species (ROS)

Neonates, particularly extremely premature infants, are repeatedly exposed to oxidative stress. The idea of oxidative stress and ROS playing significant roles in many neonatal diseases has been proposed [158,159], and new validating evidence is currently available regarding their roles in the development of NEC [160,161]. A preterm neonate’s small intestine is sensitive to hyperoxia, and excessive exposure causes barrier dysfunction, a disruption of the tight junctions [162], and a reduction in Paneth cells [26,27], leading to aggravated bacterial invasion [161]. Intestinal barrier dysfunction is a major predisposing factor in the development of NEC [163]. Oxidative stress also acts as a downstream component in the inflammatory cascade, leading to intestinal injury and eventual apoptosis [164]. Xanthine oxidase and dehydrogenase are two of the main producers of ROS in the intestines, and together with superoxides, play central roles in intestinal reperfusion injury [165,166]. Preterm infants have underdeveloped antioxidant systems, both in concentration and activity, and their ability to increase antioxidant production in response to oxidant stimuli is not present until later, in the last 15% of gestation [167]. Studies show a strong correlation between the levels of oxidative stress biomarkers in cord blood and the occurrence of NEC in preterm infants [168,169]. Excessive ROS production in a state of deficient antioxidant capacity results in mucosal injury and necrosis due to lipid membrane peroxidation or cellular protein oxidation [170]. Dietary factors also contribute to oxidative stress in preterm infants after birth. Human milk is a better scavenger of free radicals than infant formula, and less oxidative stress is demonstrated in breastfed infants [171,172,173]. ELGANs born before 28 weeks of gestation have delayed successful enteral feedings because of gut immaturity, requiring a period of dependence on total parenteral nutrition (TPN) [174]. These conditions may lead to intestinal atrophy and abnormal microbial colonization since TPN is often contaminated with oxidation products [175]. The use of bovine-based products also causes an upregulation of oxidative stress and subsequent increased intestinal permeability, and toxicity to epithelial cells. The reduced ability of intestinal epithelial cells to clear oxidative stress during enteral feedings may possibly be the initial first step in NEC pathogenesis.

## 8. Microbiome in NEC

The first colonization of normal term gut bacteria mainly comprises *Streptococcus*, *Staphyococcus*, *E. coli*, *Lactobacillus*, and *Enterobacter*, which consume oxygen for the subsequent colonization of anaerobic bacterial species, mainly *Clostridia*, *Bifidobacterium*, and members of the *Firmicutes* phylum [176]. In contrast, the microbiome of preterm infants consists of higher levels of facultative anaerobes and reduced levels of anaerobes [177], and an increased number of pathogenic bacteria such as *Escherichia coli*, *Staphylococcus*, and *Klebsiella* [178]. This perturbation, or dysbiosis, of the intestinal microbiome results in gut barrier compromise, reduced enterocyte proliferation and migration, unhindered bacterial translocation, and inappropriate and exaggerated immunologic responses [179,180,181], and it has been implicated in the development of NEC in preterm infants [182,183]. Studies show that NEC is associated with increases in *Enterobacter*, *Fusobacterium*, *Shigella*, *enterobacter sakazakii*, and *Proteobacteria,* reductions in *Bacteroides*, *Clostridium*, and *Negativicutes*, and an overall reduction in microbiome diversity [184,185]. Outbreaks of NEC were shown to be related to pathogens such as *E. coli*, *klebsiella pneumoniae*, *enterobacter cloacae*, *Salmonella*, *pseudomonas aeruginosa*, *Clostridium*, coagulase-negative staphylococci, *staphylococcus aureus*, *candida glabrata*, coronavirus, enterovirus, and rotavirus [186]. A study of 29 infants with stage 2/3 NEC found that *Klebsiella* and *Clostridium* were strongly associated with NEC [187].

## 9. Current and Future Therapies

### 9.1. Conventional Management

The lack of microbial diversity in the intestinal milieu is one of the known risk factors for the pathogenesis of NEC [188,189]. Despite advancements in understanding the pathophysiology of NEC, there has been little progress in its treatment and prevention over the past decade [190]. This may stem from the lack of a clear definition of NEC as well as from the relatively new discovery that the term “NEC”, in fact, constitutes different disease processes with a variety of pathogenic mechanisms and diagnostic and prognostic biomarkers [191]. A thorough understanding of the various entities that encompass “NEC” is therefore crucial to the successful implementation of treatment regimens targeted at the individual disease processes. Neu, J. [192] summarized some of the overlapping and differentiating factors between these different entities. Diseases that are commonly diagnosed as NEC include but are not limited to spontaneous intestinal perforation, ischemic intestinal necrosis, food protein-induced enterocolitis syndrome, and congenital gut anomalies. There is considerable overlap between the symptomology of these disease processes; however, differences in the pathophysiological mechanisms warrant more personalized therapeutic regimens. The prevention and management of “classic” NEC and the improvement in the development of beneficial gut microbiota have evolved dramatically over the past several decades. What remains unchanged is the notion of “bowel rest” that is achieved by withholding enteral feeds, gastric decompression, and a switch to parenteral nutrition. Severe cases may require an exploratory laparotomy and peritoneal drain placement [193].

### 9.2. Probiotics, Prebiotics, and Synbiotics

Advances in metagenomics have allowed for a better understanding of the neonatal gut microbiome, and it is now widely postulated that microbial dysbiosis and alterations in the gut microbiome are common predecessors of NEC. Insights into these host microbiome factors have opened an avenue of the possible uses of probiotics, prebiotics, and synbiotics as preventative and therapeutic measures to manipulate the developing microbiome and mitigate microbial dysbiosis and the prevention of NEC [194,195,196,197]. A Cochrane review of 24 randomized trials showed that probiotic preparations containing either *Lactobacillus* alone or in combination with *Bifidobacterium* reduced the incidence of severe NEC with an RR of 0.43 [198]. Another meta-analysis comprising 45 trials with 12,320 participants showed that a combination of *Bifidobacterium* and *Lactobacillus* was associated with lower rates of NEC-related morbidity [199]. Thus, the use of probiotics, prebiotics, and synbiotics as an additional prevention strategy has been adapted. Probiotics, such as *Lactobacillus* and *Bifidobacterium*, confer health benefits through multiple mechanisms, including the suppression of inflammation, upregulation of host anti-inflammatory genes, alleviation of hypoxemic injury, production of short-chain fatty acids, improved intestinal epithelial cell function, and suppression of pathogenic bacteria [195,200,201]. Prebiotics are a family of complex carbohydrates found in human breast milk that influence the microbiome [202,203,204]. They include predominantly galacto-oligosaccharides and fructo-oligosaccharides [205,206]. While some studies showed benefits for reducing the incidence of NEC, meta-analyses of randomized controlled trials showed minimal to no benefit [207,208]. Synbiotics consist of a mixture of probiotics and prebiotics that provide synergistic effects on health via their antioxidant properties that detoxify ROS [209]. However, a meta-analysis showed only low-certainty evidence about the effects of synbiotics on the risk of NEC [210].

### 9.3. Human Milk

Human breast milk is now universally accepted as the optimal starting diet for neonates and the single most important preventative strategy for reducing the incidence and severity of NEC [211,212]. It prevents dysbiosis, promotes the colonization of commensal microorganisms, and reduces the incidence of NEC [213]. The World Health Organization (WHO) recommends exclusive breastfeeding for the first six months of life [214]. Numerous studies show that the strategy of feeding human milk alone to infants decreases surgical NEC by about 90% and decreases medical NEC by 50%, partly because of its oxidative stress resistance and better antioxidant properties [215]. One of the major components of breast milk shown to prevent NEC and preserve microbiome integrity is HMOs, a category of unconjugated, multifunctional, nondigestible, and structurally diverse glycans that are unique to humans [216]. HMOs comprise about 20% of the total carbohydrates in breast milk and appear to be important substrates for the growth of commensal microorganisms such as *Bifidobacteria* [217]. HMOs support barrier function, promote immune development [218], shape the gut microbiome [219], and reduce the incidence and severity of NEC [206,220]. Milk banks, which are now increasing in numbers across the country, offer pasteurized human donor milk to those who are at risk, when an infant’s mother’s own milk is not available. The American Academy of Pediatrics has endorsed this practice since 2012 [221]. Experimental data, however, has shown that the pasteurization process that is currently used adversely affects the functional properties of human milk, including bioactive components, proteins, fatty acids, and antioxidants [222]. Perhaps a more selective process aimed at destroying the more common pathogens while preserving these properties is ideal.

### 9.4. Antioxidants

Direct and indirect antioxidant strategies that supplement components that boost antioxidant properties may offer protection. Enteral glutamines alone or in conjunction with arginine had favorable effects on oxidative stress, lipid peroxidation, and antioxidant enzyme levels in an experimental animal model of NEC [223]. Melatonin found in HM has been used to counteract oxidative stress injury in cases of asphyxia, RDS, and sepsis [224]. In NEC models, experimental rats treated with melatonin had similar oxidative stress profiles to the controls, suggesting that it has a role in reducing the severity of NEC [225]. In an animal mode of neonatal IH, early oral supplementation with glutathione showed significant improvements in diversity and commensal microorganisms compared to the controls [157].

### 9.5. Lactoferrin

Lactoferrin, which is present in human breast milk, has anti-inflammatory properties and has been recently shown to block intracellular ROS production in mesenchymal cells [226,227]. Some randomized controlled trials have demonstrated evidence, albeit low quality, that enteral lactoferrin supplementation is associated with a reduction in stage II and III NEC [228]. However, a meta-analysis of nine RCTs with 3515 samples did not show an association between enteral lactoferrin and a decrease in the incidence of NEC or all-cause mortality [229].

### 9.6. Restricted Antibiotics

A number of studies show that postnatal antibiotics decrease the diversity of the neonate’s microbiome [138,230]. A study of over 4000 extremely low-birth-weight infants who received early antibiotics showed an increased risk for NEC or death with prolonged use [231]. These and other reports suggest that the judicious use of early antibiotics in ELBW infants may be warranted. Close to 90% of extremely low-birth-weight newborns are given ampicillin, gentamicin, and/or other antibiotics despite a relatively low incidence of culture-positive early-onset sepsis between 0.2 and 0.6% [232]. Withholding antibiotics for suspected early-onset sepsis did not lead to a significant increase in neonatal mortality or morbidity in a study on routine early antibiotic use in symptomatic neonates, which was the first randomized trial in symptomatic preterm newborns to receive or not to receive antibiotics soon after birth [233]. Moreover, stopping antibiotic use as soon as the blood culture is reported to be negative at 36 to 48 h may also be implemented, and this practice may allow for the recovery of the gut microbiome diversity. Antibiotic use 48 h after birth may not have a lasting effect on the development of the gut microbiome diversity over time, and the gut microbiome diversity is recoverable [234]. The ongoing NICU Antibiotics and Outcomes (NANO) trial is currently testing the safety of withholding antibiotics as an empirical approach in suspected early neonatal sepsis and will most probably influence the excessive use of antibiotics in newborns [235]. Broad-spectrum antibiotics also remain the mainstay of treatment. No evidence for a specific choice of antibiotics or duration or frequency of treatment has been found in the literature [236].

### 9.7. Fecal Transplantation

Fecal microbiota transplantation (FMT) has also been used as a strategy for mitigating gut dysbiosis. It has been proven to effectively balance the intestinal microflora and prevent NEC in animal models [237]. Researchers have also suggested that FMT may suppress intestinal apoptosis, enhance intestinal barrier function, and decrease bacterial translocation, thereby serving as a potential novel treatment for NEC [238]. In a mouse model, FMT was shown to modulate oxidative stress and reduce colonic inflammation [239], as well as promote nitric oxide (NO) by eliminating superoxide production [240]. However, FMT may expose the recipient gut to pathogenic bacteria, and higher levels of *Escherichia* and *Salmonella enterica* have been found in FMT recipients [241]. FMT infiltration and sterilization via ultraviolet radiation can be used to sterilize fecal filtrates prior to transplantation to minimize the risk of infection [237,242]. Notably, most studies on the role of FMT in NEC are experimental, and there is a need for further research in this unexplored domain.

### 9.8. Immunotherapy

It has been established that TLR-4 is highly implicated in the pathophysiology of NEC. TLR-4 recognizes lipopolysaccharides (LPS) on Gram-negative bacteria, off-setting a cascade of pro-inflammatory signaling in the intestinal epithelium and leading to NEC [243]. TLR-4-targeted agents are being studied as potential treatment modalities for NEC. Agents that have been studied include lithocholic acid, a pregnane X receptor (PXR) agonist that inhibits TLR-4 signal expression, and glycyrrhizin, an HMBG-1 inhibitor that inactivates TLR-4 and nuclear factor kappa B (NFkB) signaling [244,245]. IL-1 receptor-associated kinase (IRAK) inhibitors have also shown to downregulate TLR-4 expression [246]. The use of TLR-4-targeted agents may prove to be an innovative treatment for NEC, but their use is limited by a lack of research on human subjects. Future prospective cohort studies should focus on understanding the mechanism of TLR-targeted therapy as well as other biological agents that can be used in the prevention and treatment of NEC.

### 9.9. Stem Cell Therapy

Stem cell therapy is increasingly being proposed as a novel therapeutic modality for NEC. It has been postulated that stem cells may contribute directly to the repair and regeneration of the intestinal epithelium owing to their unique ability to migrate to damaged tissues [247]. The mechanism of this migration is not fully understood, but may be attributed to the expression of specific receptors or ligands at the sites of inflammation and injury that facilitate multipotent stem cell infiltration and subsequent differentiation into cells of various types. There is a growing body of experimental evidence on the therapeutic effectiveness of stem cells and stem-cell-derived products in the treatment of NEC [248,249]. Bone-marrow-derived mesenchymal stem cells (BM-MSCs) have been shown to decrease intestinal inflammation and promote tissue regeneration when administered intraperitoneally to rat models. The intravenous stem cell delivery of BM-MSCs has been shown to be a less invasive yet effective method of stem cell delivery [250]. Amniotic-fluid-derived mesenchymal stem cells (AF-MSCs) have also sparked the interest of many researchers. They can be readily cultured from amniotic fluid, and when introduced intraperitoneally, have been shown to target the wingless integration (WNT)-B signaling pathway, thereby promoting cellular proliferation and epithelial regeneration. The use of other stem cell sources such as embryonic stem cells, enteral neural stem cells, and umbilical cord-derived stem cells are also promising therapeutic strategies. Similarly, studies on animal models have demonstrated that stem-cell-derived exomes may offer a therapeutic benefit. AF-MSC-derived exomes have been shown to target the WNT pathway and catenin signaling pathways, while BM-MSC-derived exomes can potentially maintain intestinal integrity in animal models, thereby proving a promising strategy in the prevention of NEC [251]. However, despite the preventative and therapeutic benefits of stem cells and exomes that were demonstrated in experimental studies on animal models, there are currently no ongoing clinical trials in human subjects. The implementation of stem cell therapy in the clinical setting is limited by concerns regarding immunological rejection, gene mutation, carcinogenesis, as well as ethical issues. In contrast, exon therapy poses a lower risk of gene mutation but can be challenging to deliver due to limitations in exome extraction and concentration that may affect treatment efficacy [252].

## 10. Conclusions

Ensuring the unperturbed establishment and development of the gut microbiota in a newborn promotes neonatal wellness, with long-term effects on the brain and immune system development. Management techniques such as early colostrum administration followed by the infant’s mother’s own breast milk; the availability of human donor milk; the widespread use of standardized feeding guidelines; the use of probiotics; and better antibiotic stewardship can contribute to promoting a healthy neonatal gut microbial environment. However, one of the challenges that remain is the prevention of and reduction in the severity of NEC. The now well-recognized damaging roles of ROS have paved the way for more research to better manage neonatal IH and oxygen therapy to mitigate ROS production. The future is wide open for innovative approaches, but nothing would be more realistic and promising than interventions directed at boosting endogenous and/or exogenous antioxidant supplementation that may not only help with prevention, but may also lessen the severity or shorten the course of the disease. Studies in our laboratory have shown tremendous synergistic benefits of combined antioxidants and fish oil for increasing the variety and abundance of commensal microorganisms and reducing the characteristics of NEC in a neonatal rat model [157]. The exclusive use of an infant’s mother’s own milk or pasteurized human donor milk, standardized feeding protocols, and the use of probiotics in very-low-birth-weight infants have decreased the incidence substantially, but because of its multifactorial nature and incompletely understood pathogenesis, NEC and its effects on the developing gut microbiome still remains difficult to eradicate.

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
