# Peer review of "Factors Influencing Neonatal Gut Microbiome and Health with a Focus on Necrotizing Enterocolitis"

_microorganisms, 2023, doi:10.3390/microorganisms11102528_

Round 1
Reviewer 1 Report
Manuscript of the overall basic factors influencing colonization of the neonatal gut. However, there are a few notes.
The provision on the colonization of the neonatal intestine by bacteria remains a controversial issue (Kennedy et al. Questioning the fetal microbiome illustrates pitfalls of low-biomass microbial studies. Nature. 2023 Jan;613(7945):639-649. doi: 10.1038/s41586-022-05546-8.)
The authors discuss the neonatal microbiome. At the same time, there is not its clear description. The microbiome of the infant goes through several stages of development, which would be better to discuss using the example of children who have had vaginal delivery, breastfeeding and the absence of antibiotics, i.e. on the example of the norm. In this case, we will see the predominance of bifidobacteria and lactobacilli after the first colonizers - facultative aerobe in the case of mothers with secretor status, because this predominance is supported by fucosylated HMOs. The predominant metabolites of this microbiota will be acetate and lactate. Complementary foods not only change the microbial composition, but also the ratio SCFAs. Following this, a discussion of microbiota's functions and the factors influencing colonization of the gut and their consequences will be more clearly understood. The detailed discussion of the different types of cell death does not make much sense for this review. At the same time, since NEC often develops in preterm newborns, it would be add the discussing of the composition of the mother's vaginal microbiota, which can provoke preterm birth. The abbreviation of the word "species" – "spp." should not italicized. The use of italicization for several parts or words in sentences in the text is also often unreasonable. The article mainly discusses bacteria, and in this case it is better to write “bacteria” or “microorganisms”, not “organisms”. It is necessary to unify the use of the Caesarean section abbreviation. The authors probably missed two references (lines 403 and 538). The manuscript has some misprints (eg. brackets - line 406, 615; Enterobacte – line 543; SFA; Breast milk is mostly composed of composed of). In case “Breast milk is mostly composed of” it would be more correct to say “Breast milk contains mostly”, since it acts as a source of microorganisms.
Author Response
Reviewer #1:
- We thank the reviewer for their suggestion to include the stages of microbiome development. Although the focus of this review is the neonatal microbiome, we have revised description of the neonatal microbiome based on the reviewer’s comments.
- The original purpose for the section on “Cell Death” was to describe how cells in the gut may undergo irreversible damage in NEC. The section has been deleted as suggested.
- Maternal vaginal microbiota composition which may provoke preterm birth has been addressed in section 7.8, as the reviewer suggested.
- We apologize for the grammatical errors and have corrected the abbreviations for spp.
- We apologize for the inadvertent italicization of certain texts. This has been corrected.
- We agree with the reviewer and have revised the “organisms” to microorganisms”.
- We have corrected the abbreviation of Cesarean section so that it is consistent throughout the text.
- We have corrected the references on lines 403 and 538.
- We have corrected the misprints on lines 406, 543, and 615.
- We have corrected the sentence regarding “breast milk composition”, according to the reviewer’s suggestion.
Reviewer 2 Report
This manuscript reviewed the factors influencing neonatal gut microbiome and health, which may lead to dysbiosis and NEC in preterm infants. The selected topic of this manuscript is meaningful and interesting. However, there are several suggestions for the author
1、In the Abstract, it is suggested to add a part of a conclusion of the whole paper.
2、In the part of Introduction, the author need to divide it into several parts according the themes of background.
3、The part of Cell Death is suggested to reduce. It seems that the relationships between the death mechanisms and microbiome is more important.
Author Response
Reviewer #2:
- We thank the reviewer for their suggestions to improve the manuscript. As suggested, we have revised the Abstract conclusion.
- The introduction has been revised and shortened as suggested.
- We have removed the section on “Cell Death”.
Round 2
Reviewer 1 Report
Please check the correct spelling of the taxonomic names of bacteria and the italics according to the recommendations (https://journals.asm.org/nomenclature, https://imafungus.biomedcentral.com/ articles/10.1186/s43008-020-00048-6#Sec2), for example, in lines 58, 332, 339, 342, 348, 394, 398). Lachnospiraceae is a family, not a genus, and the insertion of spp. in this case is an error (313). In some cases, when you discibe listing bacterial genera, it is better to write “representatives of such and such genera,” rather than indicate the generic name and species abbreviation (spp.); difference in their descriptions in text makes a bad impression. It would be good if one of the microbiologists checked the text again. Structuring into paragraphs 3-5 seems unnecessary, since the information in them is repeated in one way or another; it would be better to systematize it and present it in one paragraph. Antimicrobial peptides-secreting Paneth cells, the words “peptides” are clearly missing here: antimicrobial peptides-secreting Paneth (95). What components of vitamin B are we talking about, maybe a group of vitamins B? (241). WNT - please give the full name of the signaling pathway, this is an abbreviation. Please check the entire text again, , paying attention to the font size
Author Response
- Based on the NCBI Taxonomy Browser search engine and NCBI approved list of bacterial names, all Phyla, Order and Class are listed with the first letter capitalized and not italicized, all Family and Genus are listed with the first letter capitalized and italicized, and all Species are lower case and italicized.
- “Spp.” has been deleted when listing all Family and Genus.
- We have removed the extra paragraphs in a section.
- We have added “peptides”, as suggested.
- The components of vitamin B has been added.
- The full name of “WNT” has been spelled out.
- The font size of Palatino Linotype 9 was determined by the Microorganisms journal template, which we used for this submission.